# Asymmetric Multi-View Clustering with Hyperbolic Uncertainty Modeling

Yiming Wang [1 2]  Qun Li [1 2]  Dongxia Chang [3 4]  Jie Wen [5]  Hua Dai [1 2]  Fu Xiao [1 2]

## Abstract

Deep Multi-View Clustering (MVC) aims to learn a unified semantic representation from diverse data sources without supervision. However, current approaches relying on flat Euclidean embeddings often fail to model data uncertainty, resulting in rigid alignment where high-quality views are forced to drift toward corrupted ones. To address these challenges, we propose the Hyperbolic Asymmetric Multi-view Clustering (HAMC) framework. HAMC maps view-specific features into the Poincaré ball and uses radial geometry as a confidence proxy, encouraging confident representations to occupy larger radial distances while allowing ambiguous or noisy samples to remain closer to the origin. To mitigate noise, we introduce an asymmetric view alignment mechanism, enabling reliable views to unidirectionally guide unreliable ones. Furthermore, a consensus-aware cluster learning strategy is designed to construct robust global pseudo-labels via a confidence-based screening scheme, refining the cluster structure. Extensive experiments against 13 baselines demonstrate that HAMC achieves state-of-the-art performance.

## 1. Introduction

The rapid growth of multimodal data has transformed machine perception, driving the transition from single-source analysis to comprehensive multi-view learning (Baltrušaitis et al., 2018). From robotic navigation fusing multiple sensors (Mou et al., 2024) to multimodal medical diagno-

---

[1]School of Computer Science, Nanjing University of Posts and Telecommunications [2]State Key Laboratory of Tibetan Intelligence, Nanjing University of Posts and Telecommunications [3]Institute of Information Science, Beijing Jiaotong University [4]Visual Intelligence +X International Cooperation Joint Laboratory of MOE, Beijing Jiaotong University [5]School of Computer Science and Technology, Harbin Institute of Technology. Correspondence to: Fu Xiao <xiaof@njupt.edu.cn>.

*Proceedings of the 43rd International Conference on Machine Learning*, Seoul, South Korea. PMLR 306, 2026. Copyright 2026 by the author(s).

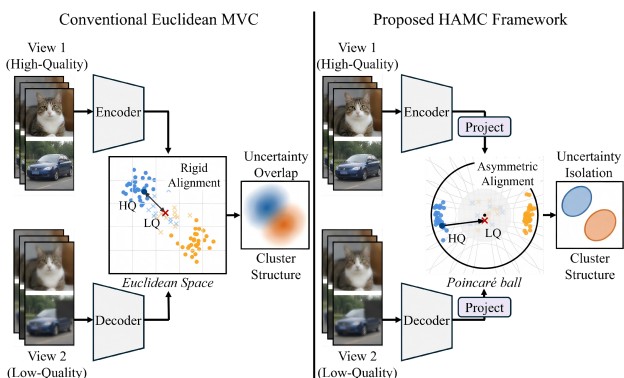

*Figure 1.* Motivation for the proposed HAMC framework. The left panel illustrates conventional Deep MVC in Euclidean space $\mathbb{R}^d$. Due to geometric mismatch, rigid symmetric alignment ($\leftrightarrow$) forces high-quality (HQ) embeddings to drift toward low-quality (LQ) views, degrading the learned representation. The right panel presents the proposed HAMC in the Poincaré ball ($\mathbb{D}^d$). The Poincaré ball provides a radial geometry for confidence modeling, where the proposed objectives encourage reliable representations to have larger radial distances and allow ambiguous or noisy representations to stay closer to the origin. This enables asymmetric guidance ($\rightarrow$), where high-confidence views unidirectionally guide less reliable ones.

sis (Kline et al., 2022), the core challenge lies in handling data heterogeneity by extracting a unified semantic consensus from diverse multi-view information. Multi-View Clustering (MVC) aims to organize this heterogeneous data into meaningful clusters without supervision. The primary goal is to identify common underlying patterns across different views while preserving unique details from each source (Fang et al., 2023; Liu et al., 2023; 2025b). This task is particularly difficult because real-world data often contains noise and lacks structured labels (Wen et al., 2022).

The development of MVC has progressed from shallow statistical models to deep learning approaches. Traditional approaches, such as graph-based (Wang et al., 2019; Tang et al., 2020; Li et al., 2021) and subspace learning methods (Sun et al., 2021; Huang et al., 2022; Liu et al., 2025a), establish the theoretical groundwork. However, they are often limited by high computational complexity and an insufficient ability to model complex nonlinear relationships. To address these bottlenecks, Deep MVC methods (Wang et al., 2023b;a; 2026) have emerged as the dominant paradigm.

By leveraging autoencoders or contrastive learning frameworks, these methods have achieved remarkable success in extracting high-level semantic features and learning nonlinear mappings (Trosten et al., 2021; Huang et al., 2024).

Despite these advancements, current deep multi-view clustering methods face two fundamental bottlenecks. First, there is a mismatch between the intrinsic complexity of data and the embedding geometry. Real-world data often exhibits varying uncertainty, leading to significant quality differences across views. Most existing methods map these complex structures into Euclidean space, which possesses a uniform structure and limited capacity (Nickel & Kiela, 2017; Ganea et al., 2018; Peng et al., 2021). This limitation hinders the model from allocating sufficient space to separate distinct clusters. Simultaneously, it struggles to handle ambiguous samples, often causing noisy features to distort the cluster structure (Chami et al., 2020; Khrulkov et al., 2020; Lin et al., 2023a). Second, standard contrastive learning minimizes the distance between view pairs regardless of their quality. In scenarios where one view is informative while another is corrupted, such rigid alignment forces the high-quality view to drift toward the noise, degrading the learned representation (Trosten et al., 2021; Lin et al., 2023b), as illustrated in the left panel of Figure 1. Although some robust methods employ weighting strategies, they generally rely on static view-level weights and fail to account for quality differences at the instance level (Yang et al., 2023; Sun et al., 2024).

To address these challenges, we propose the Hyperbolic Asymmetric Multi-view Clustering (HAMC) framework. Multi-view features are embedded into the Poincaré ball model (Nickel & Kiela, 2017) to leverage its exponential capacity. Using the radial structure of the Poincaré ball together with entropy-aware learning objectives, HAMC encourages discriminative, high-confidence representations to occupy larger radial distances, while allowing ambiguous or noisy representations to remain closer to the origin, as depicted in the right panel of Figure 1. Specifically, we introduce an asymmetric view alignment mechanism that utilizes prediction entropy to enforce directional guidance, ensuring that high-confidence views unidirectionally guide less reliable ones. Finally, a consensus-aware cluster learning strategy refines cluster prototypes by constructing robust global pseudo-labels using reliable samples to prevent noise propagation. The main contributions of this work are summarized as follows:

- We introduce hyperbolic geometry to deep multi-view clustering to address the geometric mismatch. Leveraging the Poincaré ball, our method encourages high-confidence representations to have larger radial distances and allows ambiguous or noisy samples to stay closer to the origin.

- We propose HAMC, which replaces rigid alignment with asymmetric guidance from reliable to corrupted views based on instance-level uncertainty. Furthermore, it is integrated with a consensus-aware learning strategy that employs decision-level fusion to construct screened global pseudo-labels for robust fusion.

- Extensive experiments on diverse benchmarks demonstrate that HAMC significantly outperforms state-of-the-art methods in both accuracy and robustness.

## 2. Related Work

This section provides a brief review of some related works in the domains of deep multi-view clustering and hyperbolic representation learning.

### 2.1. Deep Multi-View Clustering

Deep multi-view clustering has evolved significantly in recent years. Early methods primarily adopted a combination of autoencoders and traditional clustering algorithms. Recently, contrastive learning has become the dominant paradigm. It aims to learn a common semantic representation by maximizing the mutual information between consistent views (Fang et al., 2023; Wang et al., 2025). Beyond simple pairwise alignment, recent research focuses on capturing global structural and semantic consistencies. For instance, Chen et al. (Chen et al., 2023) refine semantic alignment by contrasting cluster assignments to filter inconsistent information. Similarly, high-order random walks are integrated in (Lu et al., 2024) to enhance feature distinctiveness and extract semantic invariant features. Addressing missing modalities is another critical challenge. Lin et al. (Lin et al., 2023b) pioneer dual contrastive prediction to recover missing views by minimizing conditional entropy. From a prototype perspective, methods such as (Jin et al., 2023) and (Li et al., 2025) utilize cluster prototypes to infer missing views or align partial samples directly.

Despite these diverse improvements, a critical bottleneck remains. Most methods rely on symmetric alignment strategies within Euclidean space. They typically enforce rigid alignment by minimizing distances between views regardless of their instance-level quality or uncertainty. When a view is corrupted or uninformative, this rigid mechanism forces high-quality representations to drift toward noise, inevitably degrading the cluster structure. Unlike these approaches, our HAMC introduces an asymmetric alignment mechanism, allowing reliable views to unidirectionally guide unreliable ones, thereby preventing noise propagation.

### 2.2. Hyperbolic Representation Learning

Representation learning typically relies on Euclidean space. However, Euclidean geometry is uniform and has limited ca-

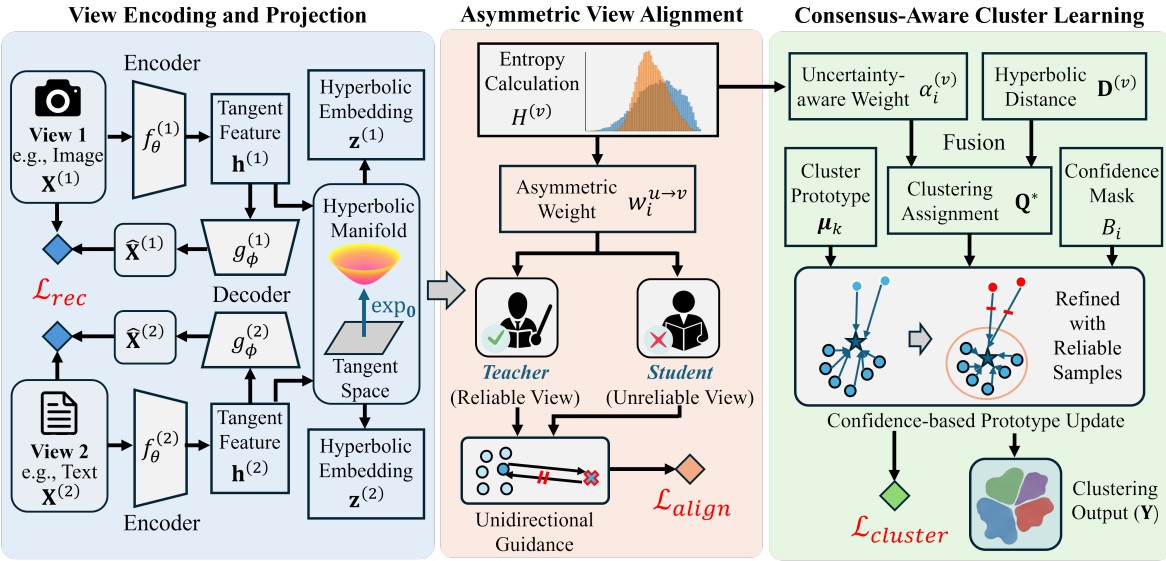

*Figure 2.* The overall framework of Hyperbolic Asymmetric Multi-view Clustering (HAMC).

pacity (Peng et al., 2021). These characteristics make it difficult to embed complex data structures or represent varying levels of uncertainty. In contrast, hyperbolic geometry offers an exponential expansion of volume. This unique property makes it ideal for modeling data with latent hierarchical structures and encoding uncertainty. Seminal works such as Poincaré embeddings (Nickel & Kiela, 2017) and hyperbolic neural networks (Ganea et al., 2018) have demonstrated the superiority of hyperbolic space in natural language processing and graph learning. Recently, hyperbolic geometry has also been applied to computer vision tasks, such as image segmentation (Atigh et al., 2022) and image retrieval (Desai et al., 2023). Additionally, researchers (Chami et al., 2020; Long & van Noord, 2023) have applied it to hierarchical clustering to handle large-scale data.

However, its potential in Deep MVC remains largely unexplored. Most existing MVC methods still map complex multi-view features into flat Euclidean space. This leads to a geometric mismatch, as the fixed curvature of Euclidean space cannot handle the varying uncertainty inherent in multi-view data. Specifically, Euclidean space suffers from limited boundary capacity, forcing discriminative features to overlap with noise. Our work bridges this gap by using the radial structure of the Poincaré ball as a confidence proxy, so that high-confidence samples can be represented with larger radial distances while noisy samples are allowed to stay closer to the origin under the proposed objectives.

## 3. Methodology

In this section, we detail the proposed Hyperbolic Asymmetric Multi-view Clustering (HAMC) framework. As illustrated in Figure 2, HAMC consists of three modules: View Encoding and Projection, Asymmetric View Alignment, and Consensus-Aware Cluster Learning. We use the embedding norm as a confidence-related proxy, which supports an asymmetric view alignment mechanism for directional guidance and a consensus-aware clustering strategy that selectively utilizes reliable samples for prototype refinement.

### 3.1. Preliminaries

**Problem Definition.** Consider a multi-view dataset $\mathcal{X} = \{\mathbf{x}_i\}_{i=1}^N$ with $N$ samples. Each sample $\mathbf{x}_i$ consists of $M$ views, denoted as $\{\mathbf{x}_i^{(1)}, \mathbf{x}_i^{(2)}, \ldots, \mathbf{x}_i^{(M)}\}$, where $\mathbf{x}_i^{(v)} \in \mathbb{R}^{D_v}$ represents the raw feature of the $v$-th view with dimension $D_v$. Our goal is to jointly learn a shared latent representation space and a set of cluster prototypes $\boldsymbol{\mu} = \{\boldsymbol{\mu}_k\}_{k=1}^K$, where $\boldsymbol{\mu}_k$ represents the semantic center of the $k$-th cluster.

**The Poincaré Ball Model of Hyperbolic Geometry.** Standard multi-view clustering methods typically operate in Euclidean space. However, views often exhibit significant discrepancies in information density and uncertainty. Euclidean geometry fails to represent these differences effectively due to its flat curvature and polynomial volume growth. To address this mismatch, we ground our framework in hyperbolic geometry, a non-Euclidean geometry with constant negative curvature.

Formally, a $d$-dimensional Riemannian manifold can be defined as a pair $(\mathcal{M}, g)$. At any point $\mathbf{z}$, it is locally approximated by a linear Euclidean space $\mathbb{R}^d$, referred to as the tangent space $\mathcal{T}_\mathbf{z}\mathcal{M}$. In this work, we adopt the $d$-dimensional Poincaré ball model $(\mathbb{D}^d, g^\mathbb{D})$ as the realization of hyperbolic space. It is defined as an open ball

$\mathbb{D}^d = \{\mathbf{z} \in \mathbb{R}^d : \|\mathbf{z}\|_2 < 1\}$ with curvature $-1$. The Riemannian metric $g_{\mathbf{z}}^{\mathbb{D}}$ at point $\mathbf{z}$ is conformal to the Euclidean metric $g^E$, defined as: $g_{\mathbf{z}}^{\mathbb{D}} = \lambda_{\mathbf{z}}^2 g^E$, where $\lambda_{\mathbf{z}} = \frac{2}{1 - \|\mathbf{z}\|_2^2}$ is the conformal factor. This metric implies an exponential expansion of volume near the boundary $\partial \mathbb{D}$, providing larger geometric capacity for separating distinct clusters. Since $d_{\mathbb{D}}(\mathbf{0}, \mathbf{z}) = 2 \operatorname{arctanh}(\|\mathbf{z}\|_2)$ is monotonic with $\|\mathbf{z}\|_2$, we use radial position as a confidence proxy rather than assuming that noisy samples are automatically pushed to the origin (Khrulkov et al., 2020).

To project a Euclidean feature onto this manifold, we utilize the exponential map $\exp_{\mathbf{0}} : \mathcal{T}_{\mathbf{0}} \mathbb{D}^d \to \mathbb{D}^d$. For a vector $\mathbf{h} \in \mathbb{R}^d$ in the tangent space at the origin, it is defined as:

$$\exp_{\mathbf{0}}(\mathbf{h}) = \begin{cases} \tanh(\|\mathbf{h}\|_2) \dfrac{\mathbf{h}}{\|\mathbf{h}\|_2}, & \mathbf{h} \neq \mathbf{0}, \\ \mathbf{0}, & \mathbf{h} = \mathbf{0}. \end{cases} \tag{1}$$

With data projected onto the manifold, the hyperbolic distance between two points $\mathbf{z}_i, \mathbf{z}_j \in \mathbb{D}^d$ is derived as:

$$d_{\mathbb{D}}(\mathbf{z}_i, \mathbf{z}_j) = \operatorname{arccosh}\left(1 + \frac{2\|\mathbf{z}_i - \mathbf{z}_j\|_2^2}{(1 - \|\mathbf{z}_i\|_2^2)(1 - \|\mathbf{z}_j\|_2^2)}\right). \tag{2}$$

This metric serves as the basis for our subsequent similarity measurement and prototype updating, ensuring that geometric properties are preserved during the clustering process.

### 3.2. View Encoding and Projection

To effectively capture semantic information from diverse views while explicitly modeling their uncertainty, we employ a hyperbolic autoencoder framework. This module consists of a hyperbolic mapping layer and a set of view-specific encoders and decoders.

For the $v$-th view of the $i$-th sample $\mathbf{x}_i^{(v)}$, the encoder $f_\theta^{(v)}$ first extracts a latent feature. Since neural networks naturally operate in Euclidean space, the output of this encoder is regarded as a vector in the tangent space $\mathcal{T}_{\mathbf{0}} \mathbb{D}^d$ at the origin. This tangent feature $\mathbf{h}_i^{(v)} \in \mathbb{R}^d$ is formalized as:

$$\mathbf{h}_i^{(v)} = f_\theta^{(v)}(\mathbf{x}_i^{(v)}), \tag{3}$$

where $\theta$ denotes the learnable parameters of the encoder.

Subsequently, we project $\mathbf{h}_i^{(v)}$ onto the Poincaré ball manifold $\mathbb{D}^d$ using the exponential map defined in Preliminaries. The hyperbolic embedding $\mathbf{z}_i^{(v)}$ is computed as $\mathbf{z}_i^{(v)} = \exp_{\mathbf{0}}(\mathbf{h}_i^{(v)})$. Through this projection, the radial norm $\|\mathbf{z}_i^{(v)}\|_2$ serves as a confidence-related geometric quantity. Since radial distance in the Poincaré ball is monotonic with the distance from the origin, larger radial positions provide more capacity for separating confident representations.

Under the subsequent entropy-aware alignment and clustering objectives, confident samples are encouraged to obtain larger radial distances, while ambiguous or noisy samples can remain closer to the origin.

To ensure training stability and avoid the numerical instability often associated with hyperbolic optimization, we employ a tangent space decoding strategy. Instead of decoding from the hyperbolic embedding $\mathbf{z}_i^{(v)}$, the decoder $g_\phi^{(v)}$ reconstructs the input directly from the tangent feature $\mathbf{h}_i^{(v)}$:

$$\hat{\mathbf{x}}_i^{(v)} = g_\phi^{(v)}(\mathbf{h}_i^{(v)}). \tag{4}$$

The reconstruction loss is then minimized using the Mean Squared Error (MSE):

$$\mathcal{L}_{rec} = \sum_{v=1}^{M} \sum_{i=1}^{N} \|\mathbf{x}_i^{(v)} - \hat{\mathbf{x}}_i^{(v)}\|_2^2. \tag{5}$$

This strategy effectively decouples the geometric constraints from feature retention. It leverages the stability of Euclidean operations for reconstruction while reserving the hyperbolic space specifically for uncertainty modeling and clustering.

### 3.3. Asymmetric View Alignment

Traditional multi-view clustering often enforces rigid symmetric alignment, forcing embeddings from different views to be close regardless of their quality. This can be detrimental when one view is corrupted or uninformative. To address this, we propose the asymmetric view alignment mechanism. This module leverages cluster prototypes to estimate instance-level uncertainty and imposes a directional alignment constraint, ensuring that high-confidence views unidirectionally guide low-confidence ones.

To implement this, we introduce a set of learnable cluster prototypes $\boldsymbol{\mu} = \{\boldsymbol{\mu}_k\}_{k=1}^{K}$ to represent the semantic centers of the $K$ clusters. Consistent with our tangent space decoding strategy, these prototypes are maintained in the tangent space as $\{\mathbf{p}_k\}_{k=1}^{K} \subset \mathbb{R}^d$. Then, we project them onto the Poincaré ball $\mathbb{D}^d$ via the exponential map for distance computation: $\boldsymbol{\mu}_k = \exp_{\mathbf{0}}(\mathbf{p}_k)$. This design enables numerical stability during iterative updates.

For the $v$-th view, we first calculate the hyperbolic distance matrix $\mathbf{D}^{(v)} \in \mathbb{R}^{N \times K}$ between embeddings and prototypes, where $D_{ik}^{(v)} = d_{\mathbb{D}}(\mathbf{z}_i^{(v)}, \boldsymbol{\mu}_k)$. To obtain a robust soft assignment probability $\mathbf{Q}^{(v)} \in \mathbb{R}^{N \times K}$, we treat $-\mathbf{D}^{(v)}$ as assignment logits, convert it into the non-negative kernel $\exp(-\mathbf{D}^{(v)})$, and apply the Sinkhorn-Knopp algorithm (Cuturi, 2013; Caron et al., 2020) to obtain $\mathbf{Q}^{(v)}$.

The probability $Q_{ik}^{(v)}$ represents the likelihood of sample $i$ belonging to cluster $k$ in view $v$. Subsequently, we quantify the uncertainty of sample $i$ in view $v$ using the predictive

entropy $H_i^{(v)}$:

$$H_i^{(v)} = -\sum_{k=1}^{K} Q_{ik}^{(v)} \log \left( Q_{ik}^{(v)} + \epsilon \right), \qquad (6)$$

where $\epsilon$ is a small constant for numerical stability. This entropy is used as a prediction-based uncertainty measure associated with the current hyperbolic prototype assignment. In the Poincaré ball, radial distance affects distances to prototypes and thus can influence the sharpness of the assignment distribution. Samples with clearer prototype preference usually yield lower entropy, whereas ambiguous samples tend to produce more diffuse assignments and higher entropy. Therefore, entropy provides an operational estimate of instance-level uncertainty rather than a direct consequence of geometry alone.

Based on the estimated entropy, we construct a dynamic gating mechanism to regulate the information flow. We define an asymmetric weight $w_i^{u\to v}$ that activates only when the source view $u$ is more confident than the target view $v$:

$$w_i^{u\to v} = \text{sg} \left( \max(0, H_i^{(v)} - H_i^{(u)}) \right). \qquad (7)$$

This weight determines the strength of the guidance. If view $u$ exhibits lower entropy and higher confidence, it acts as a teacher to guide view $v$. Otherwise, the weight becomes zero. This mechanism effectively prevents uncertain views from damaging the learned cluster structure. Consequently, we formulate the alignment loss using hyperbolic distance:

$$\mathcal{L}_{align} =$$
$$\frac{1}{NM(M-1)} \sum_{i=1}^{N} \sum_{u=1}^{M} \sum_{\substack{v=1 \\ v\neq u}}^{M} w_i^{u\to v} d_{\mathbb{D}} \left( \text{sg}(\mathbf{z}_i^{(u)}), \mathbf{z}_i^{(v)} \right)^2,$$
$$(8)$$

where $\text{sg}(\cdot)$ denotes the stop-gradient operation. In Eq. 7, the asymmetric weight is detached from the computational graph. In Eq. 8, the source embedding is also detached. Therefore, for an active ordered pair where view $u$ has lower entropy than view $v$, the alignment loss updates the less reliable target view $v$ without directly pulling the more reliable source view $u$ toward it.

### 3.4. Consensus-Aware Cluster Learning

Although the asymmetric alignment fosters consistency between view pairs, relying solely on local view predictions fails to exploit the global semantic structure. To establish reliable clustering targets, we propose a consensus-aware cluster learning strategy. This module operates on two distinct levels: First, for each sample, we integrate multi-view predictions via an uncertainty-aware decision fusion mechanism to construct a robust global consensus. Second, across

the entire dataset, we employ a confidence-based screening strategy to filter out unreliable samples, ensuring that only reliable samples are utilized for representation learning and prototype update.

For instance-level prediction fusion, instead of treating all views equally, we integrate prediction confidence to form a unified consensus logit. We compute an uncertainty-aware weight $\alpha_i^{(v)}$ for each view based on the previously calculated entropy $H_i^{(v)}$:

$$\alpha_i^{(v)} = \frac{\exp(-H_i^{(v)}/\tau)}{\sum_{u=1}^{M} \exp(-H_i^{(u)}/\tau)}, \qquad (9)$$

where $\tau$ is a temperature parameter. Unlike the pairwise weight in the alignment module, this weight $\alpha_i^{(v)}$ reflects the contribution of view $v$ to the global consensus. We then aggregate the negative distance matrices $-\mathbf{D}^{(v)}$ to compute the weighted consensus logits $\bar{\mathbf{S}} \in \mathbb{R}^{N \times K}$:

$$\bar{S}_{ik} = \sum_{v=1}^{M} \alpha_i^{(v)}(-D_{ik}^{(v)}). \qquad (10)$$

To ensure the global partition satisfies balance assignment constraints, we treat $\bar{\mathbf{S}}$ as consensus logits and apply the Sinkhorn-Knopp algorithm to the non-negative kernel $\exp(\bar{\mathbf{S}})$ to obtain the optimized global assignment $\mathbf{Q}^*$. The final pseudo-label for sample $i$ is determined by $y_i^* = \arg\max_k Q_{ik}^*$.

On the dataset level, to prevent noise propagation from low-quality samples, we adopt a confidence-based screening strategy to utilize only reliable samples. Specifically, we introduce a binary mask $B_i \in \{0, 1\}$ that selects only the top-$p$ most confident samples based on the maximum probability in $\mathbf{Q}^*$. The clustering loss is defined as the hyperbolic cross-entropy between the view-specific embeddings and the global pseudo-labels:

$$\mathcal{L}_{cluster} = -\frac{1}{M\sum_{i=1}^{N} B_i} \sum_{i=1}^{N} B_i \sum_{v=1}^{M}$$
$$\cdot \log \frac{\exp(-d_{\mathbb{D}}(\mathbf{z}_i^{(v)}, \boldsymbol{\mu}_{y_i^*})^2/\tau_c)}{\sum_{k=1}^{K} \exp(-d_{\mathbb{D}}(\mathbf{z}_i^{(v)}, \boldsymbol{\mu}_k)^2/\tau_c)}, \qquad (11)$$

where $\tau_c$ is the clustering temperature. This objective encourages view-specific embeddings to align with the high-confidence global consensus, fostering a unified semantic representation across all views.

Finally, we update the cluster prototypes using the screened reliable samples. Since hyperbolic space optimization can be unstable, we perform updates in the tangent space. Specifically, for each cluster $k$, we compute a candidate center $\mathbf{c}_k$ in the tangent space by aggregating the features $\mathbf{h}_i^{(v)}$ of

assigned reliable samples, weighted by the view confidence $\alpha_i^{(v)}$. The prototype $\mathbf{p}_k$ is then updated via a momentum strategy:

$$\mathbf{p}_k^{(t+1)} = m\mathbf{p}_k^{(t)} + (1-m)\mathbf{c}_k, \quad (12)$$

where $m$ is the momentum coefficient. The updated tangent prototype is then projected back to the Poincaré ball as $\boldsymbol{\mu}_k = \exp_{\mathbf{0}}(\mathbf{p}_k)$.

### 3.5. Optimization Objective

The optimization of HAMC follows a two-stage strategy to ensure stable convergence and avoid degenerate solutions.

**Stage 1: Warm-up.** In the first stage, we focus solely on initializing the view encoders and decoders. By minimizing the reconstruction loss and a symmetric contrastive loss, we ensure that the tangent space features $\mathbf{h}_i^{(v)}$ capture the intrinsic data structure before clustering begins. The warm-up objective is formulated as:

$$\mathcal{L}_{warmup} = \mathcal{L}_{rec} + \lambda_{wc} \sum_{u \neq v} \mathcal{L}_{NCE}(\mathbf{h}^{(u)}, \mathbf{h}^{(v)}), \quad (13)$$

where $\lambda_{wc}$ is a weighting coefficient and $\mathcal{L}_{NCE}$ denotes the standard InfoNCE loss (Oord et al., 2018). This stage equips the tangent feature space with preliminary discriminability.

**Stage 2: Joint Training.** In the second stage, we perform end-to-end training of the entire framework. We jointly optimize the network parameters and cluster prototypes using the total objective function:

$$\mathcal{L}_{total} = \mathcal{L}_{rec} + \lambda_1 \mathcal{L}_{align} + \lambda_2 \mathcal{L}_{cluster}, \quad (14)$$

where $\lambda_1$ and $\lambda_2$ are hyper-parameters. The reconstruction loss $\mathcal{L}_{rec}$ ensures that the tangent features retain the intrinsic information of the original input, preventing degenerate solutions where embeddings collapse. The asymmetric alignment loss $\mathcal{L}_{align}$ enforces cross-view consistency by transferring structural knowledge from high-confidence views to ambiguous ones. Finally, the consensus clustering loss $\mathcal{L}_{cluster}$ optimizes the decision boundaries based on the global consensus, enhancing the discriminability of the latent space. The entire framework is trained end-to-end simultaneously, with cluster prototypes updated via the momentum strategy.

## 4. Experiments

In this section, we conduct experiments on six popular multi-view datasets to evaluate the effectiveness of HAMC.

### 4.1. Experimental Settings

**Datasets.** To evaluate the effectiveness and robustness of HAMC, we conduct experiments on six widely used multi-view datasets, including Caltech-101 (Li et al., 2015), CUB (Wah et al., 2011), LandUse-21 (Yang & Newsam, 2010), NoisyMNIST (Wang et al., 2015), Reuters (Amini et al., 2009) and Scene-15 (Fei-Fei & Perona, 2005). In the appendix, we provide detailed descriptions of these datasets.

**Baselines.** We compare HAMC with 13 state-of-the-art multi-view clustering methods, ranging from classic deep learning approaches to recent robust frameworks. Specifically, AE2-Nets (Zhang et al., 2019a) and BMVC (Zhang et al., 2019b) are designed for complete multi-view clustering. The remaining 11 baselines are designed for multi-view clustering with incomplete information, including PVC (Huang et al., 2020), EERIMVC (Liu et al., 2021), MvCLN (Yang et al., 2021), DSIMVC (Tang & Liu, 2022), DCP (Lin et al., 2023b), SURE (Yang et al., 2023), ICMVC (Chao et al., 2024), CGCN (Wang et al., 2024), CANDY (Guo et al., 2024), SMILE (Zeng et al., 2024), and CAMERA (Li et al., 2025). For evaluation, we employ three standard metrics: Clustering Accuracy (ACC), Normalized Mutual Information (NMI), and Adjusted Rand Index (ARI). For all metrics, higher values indicate better performance.

**Implementation Details.** The proposed HAMC is implemented using the PyTorch framework on an NVIDIA RTX 4090 GPU. For all datasets, we employ fully connected networks as the view-specific autoencoders. We determine the optimal hyper-parameters via grid search. Specifically, the dimension of the Poincaré ball embedding $d$ is tuned from $\{64, 128, 256\}$ to match different dataset complexities. The weighting coefficient $\lambda_{wc}$ in the warm-up stage and the joint training trade-off parameters $\lambda_1$ and $\lambda_2$ are tuned within the range of $\{10^{-3}, 10^{-2}, 10^{-1}, 1, 10, 100\}$. Additionally, the momentum coefficient $m$ is tuned within the range of $\{0.9, 0.95, 0.99\}$, while the confidence-based screening ratio $p$ is set to 0.5. The temperature parameters are set to $\tau = 0.3$ and $\tau_c = 0.7$. Furthermore, we verify the effectiveness of our method in incomplete MVC scenarios, where some views of samples may be missing during data collection. Specifically, to simulate incomplete data, we randomly select $N_{miss} = \eta \times N$ samples and remove one view, where $\eta$ denotes the missing rate.

### 4.2. Comparison with State-of-the-Art Methods

We compare HAMC with 13 baseline methods on six benchmark datasets. Table 1 reports the clustering performance under both complete and incomplete (50% missing rate) settings. The detailed analysis is summarized as follows:

- **Superior Performance:** As evidenced in Table 1, HAMC achieves the best performance on the majority of datasets across both settings. These consistent improvements demonstrate that our framework effectively addresses the geometric mismatch problem and demonstrates its effectiveness compared to existing

*Table 1.* Comparison of clustering performance on six benchmark datasets. The best results are highlighted in **bold**, and the second-best results are underlined.

| Setting | Methods | Caltech-101 | | | CUB | | | LandUse-21 | | | NoisyMNIST | | | Reuters | | | Scene-15 | | |
|---|---|---|---|---|---|---|---|---|---|---|---|---|---|---|---|---|---|---|---|
| | | ACC | NMI | ARI | ACC | NMI | ARI | ACC | NMI | ARI | ACC | NMI | ARI | ACC | NMI | ARI | ACC | NMI | ARI |
| Complete View | AE2-Nets | 7.6 | 12.5 | 0.8 | 54.3 | 49.9 | 34.9 | 24.8 | 30.4 | 10.4 | 42.1 | 43.4 | 30.4 | 42.4 | 19.8 | 14.9 | 37.2 | 40.5 | 22.2 |
| | BMVC | 50.1 | 72.4 | 33.9 | 66.2 | 61.7 | 48.7 | 25.3 | 28.6 | 11.4 | 88.3 | 77.0 | 76.6 | 42.4 | 21.9 | 15.1 | 40.5 | 41.2 | 24.1 |
| | PVC | 20.5 | 51.4 | 15.7 | 59.6 | 66.7 | 52.9 | 16.8 | 25.2 | 5.6 | 87.1 | 92.8 | 93.1 | 38.0 | 20.3 | 10.1 | 38.0 | 39.8 | 21.1 |
| | EERIMVC | 49.0 | 74.2 | 34.2 | 74.0 | 73.1 | 62.4 | 24.9 | 29.6 | 12.2 | 65.7 | 57.6 | 51.3 | 33.2 | 14.3 | 3.9 | 39.6 | 39.0 | 22.1 |
| | MvCLN | 39.6 | 65.3 | 32.8 | 64.9 | 60.0 | 47.8 | 26.1 | 30.7 | 12.5 | 97.3 | 94.2 | 95.3 | 50.6 | 29.6 | 25.7 | 37.9 | 42.3 | 25.6 |
| | DSIMVC | 21.4 | 37.0 | 17.2 | 59.1 | 57.5 | 41.2 | 18.1 | 18.6 | 5.6 | 57.1 | 54.9 | 43.6 | 43.2 | 23.3 | 19.0 | 30.7 | 35.2 | 17.1 |
| | DCP | 49.2 | 74.4 | 49.0 | 61.5 | 68.3 | 48.5 | 26.2 | 32.7 | 13.5 | 81.2 | 86.0 | 75.8 | 36.2 | 18.9 | 4.8 | 39.4 | 41.5 | 21.0 |
| | SURE | 43.8 | 70.1 | 29.5 | 62.7 | 60.1 | 46.1 | 25.1 | 28.3 | 10.9 | 98.4 | 95.4 | 96.5 | 49.1 | 29.9 | 23.6 | 42.8 | 42.5 | 24.6 |
| | ICMVC | 34.0 | 64.0 | 40.0 | 83.0 | 77.1 | 69.8 | 27.8 | 31.6 | 14.5 | 98.7 | 96.3 | 97.2 | 52.8 | 30.3 | 24.9 | 41.2 | 43.6 | 25.7 |
| | CGCN | 49.1 | 75.2 | 33.8 | 70.7 | 71.5 | 60.3 | 28.8 | 36.0 | 15.0 | 97.6 | 95.2 | 96.5 | 45.8 | 27.0 | 22.3 | 42.9 | 43.4 | 25.0 |
| | CANDY | 67.3 | 83.8 | 60.0 | 80.8 | 76.2 | 68.1 | 30.6 | 36.5 | 16.2 | 99.1 | 97.0 | 98.3 | 57.7 | 30.8 | 37.1 | 42.0 | 41.6 | 24.7 |
| | SMILE | 51.0 | 79.4 | 35.3 | 74.7 | 75.5 | 64.5 | 26.7 | 29.1 | 13.1 | 99.3 | 97.8 | 98.4 | 42.5 | 32.9 | 26.2 | 44.4 | 44.6 | 27.4 |
| | CAMERA | 65.6 | 74.8 | 51.5 | 81.4 | 75.6 | 66.6 | 28.6 | 35.4 | 15.1 | 99.9 | 99.6 | 99.7 | 55.0 | 34.8 | 29.5 | 45.5 | 46.3 | 28.2 |
| | HAMC (Ours) | 70.7 | 84.7 | 69.7 | 87.5 | 81.5 | 75.2 | 30.9 | 36.8 | 17.1 | 99.4 | 98.0 | 98.6 | 61.2 | 44.2 | 33.0 | 47.7 | 47.5 | 30.6 |
| Incomplete View with $\eta = 50\%$ | AE2-Nets | 6.6 | 18.0 | 4.5 | 35.9 | 32.0 | 15.9 | 19.2 | 23.0 | 5.8 | 29.9 | 23.8 | 11.8 | 29.1 | 7.6 | 4.8 | 22.4 | 23.4 | 9.6 |
| | BMVC | 40.0 | 58.5 | 10.2 | 29.8 | 20.3 | 6.4 | 18.8 | 18.7 | 3.7 | 30.7 | 19.2 | 10.6 | 32.1 | 7.0 | 2.9 | 32.5 | 30.9 | 11.6 |
| | PVC | 6.6 | 17.4 | 0.3 | 39.0 | 40.5 | 20.9 | 21.3 | 23.1 | 8.1 | 16.4 | 6.7 | 2.3 | 20.7 | 5.3 | 3.8 | 27.0 | 23.5 | 10.6 |
| | EERIMVC | 43.6 | 69.0 | 26.4 | 68.7 | 63.9 | 53.8 | 22.1 | 25.2 | 9.1 | 55.6 | 45.9 | 36.8 | 29.8 | 12.0 | 4.2 | 28.9 | 27.0 | 8.4 |
| | MvCLN | 27.2 | 47.5 | 23.5 | 45.2 | 40.8 | 21.9 | 22.1 | 25.2 | 9.1 | 53.8 | 50.6 | 28.5 | 39.3 | 18.4 | 14.3 | 31.4 | 29.5 | 13.9 |
| | DSIMVC | 16.4 | 24.8 | 9.2 | 54.4 | 52.4 | 35.2 | 18.6 | 18.8 | 5.7 | 55.8 | 55.1 | 43.0 | 39.9 | 19.6 | 17.1 | 30.6 | 35.5 | 17.2 |
| | DCP | 44.3 | 71.0 | 45.3 | 53.7 | 65.5 | 47.3 | 22.2 | 27.0 | 10.4 | 80.0 | 75.2 | 70.7 | 34.6 | 17.5 | 2.9 | 39.5 | 42.4 | 23.5 |
| | SURE | 34.6 | 57.8 | 19.9 | 58.3 | 50.4 | 37.4 | 23.1 | 28.6 | 10.6 | 93.0 | 85.4 | 85.9 | 47.2 | 30.9 | 23.3 | 39.6 | 41.6 | 23.5 |
| | ICMVC | 32.2 | 61.3 | 36.9 | 76.7 | 70.5 | 65.9 | 26.6 | 28.7 | 12.2 | 96.2 | 92.0 | 92.2 | 49.4 | 27.9 | 22.2 | 34.0 | 33.2 | 23.3 |
| | CGCN | 45.2 | 70.8 | 30.1 | 67.2 | 69.1 | 58.4 | 23.9 | 28.9 | 13.1 | 94.7 | 93.3 | 93.9 | 42.9 | 24.7 | 21.0 | 34.9 | 35.1 | 22.0 |
| | CANDY | 69.5 | 83.9 | 65.5 | 71.0 | 67.1 | 55.8 | 28.8 | 31.1 | 14.4 | 95.7 | 91.6 | 92.3 | 54.2 | 34.8 | 27.2 | 40.0 | 40.2 | 24.1 |
| | SMILE | 51.2 | 79.0 | 35.6 | 69.5 | 66.7 | 54.9 | 24.5 | 28.3 | 11.4 | 96.8 | 91.7 | 93.0 | 39.4 | 30.0 | 23.5 | 41.5 | 41.3 | 25.3 |
| | CAMERA | 63.8 | 71.7 | 48.2 | 74.1 | 68.1 | 56.3 | 27.2 | 31.9 | 13.4 | 98.3 | 95.1 | 96.3 | 54.4 | 35.9 | 30.3 | 44.9 | 44.4 | 26.9 |
| | HAMC (Ours) | 69.1 | 83.3 | 66.8 | 82.8 | 76.3 | 70.7 | 29.1 | 34.4 | 15.6 | 98.1 | 95.9 | 97.5 | 57.6 | 41.5 | 30.9 | 46.5 | 44.8 | 28.2 |

*Table 2.* Performance comparison of HAMC and its variants on six benchmark datasets. The best results are highlighted in **bold**.

| Variants | Caltech-101 | | | CUB | | | LandUse-21 | | | NoisyMNIST | | | Reuters | | | Scene-15 | | |
|---|---|---|---|---|---|---|---|---|---|---|---|---|---|---|---|---|---|---|
| | ACC | NMI | ARI | ACC | NMI | ARI | ACC | NMI | ARI | ACC | NMI | ARI | ACC | NMI | ARI | ACC | NMI | ARI |
| w/o Hyperbolic | 57.0 | 81.3 | 40.1 | 79.2 | 79.6 | 64.2 | 23.4 | 31.0 | 14.0 | 99.0 | 97.2 | 97.9 | 45.1 | 22.1 | 18.8 | 45.1 | 45.9 | 28.2 |
| w/o Asymmetric Align | 66.2 | 83.9 | 57.9 | 77.8 | 80.5 | 60.7 | 27.6 | 33.1 | 16.3 | 98.7 | 96.2 | 97.2 | 48.0 | 32.9 | 21.3 | 46.8 | 47.3 | 30.0 |
| w/o Uncertainty Weight | 65.7 | 84.0 | 57.6 | 82.8 | 81.3 | 68.2 | 28.7 | 32.8 | 16.1 | 99.3 | 97.8 | 98.4 | 50.2 | 34.1 | 26.5 | 46.9 | 47.2 | 30.1 |
| w/o Confidence Mask | 52.9 | 78.2 | 35.8 | 75.6 | 75.7 | 61.1 | 22.6 | 32.2 | 11.8 | 99.1 | 97.4 | 98.1 | 47.3 | 24.3 | 20.3 | 44.9 | 45.7 | 27.9 |
| **HAMC** | **70.7** | **84.7** | **69.7** | **87.5** | **81.5** | **75.2** | **30.9** | **36.8** | **17.1** | **99.4** | **98.0** | **98.6** | **61.2** | **44.2** | **33.0** | **47.7** | **47.5** | **30.6** |

approaches.

- **Effectiveness on Complete Data:** On complete-view datasets, HAMC shows clear advantages on CUB and achieves strong performance on Reuters. On CUB, it outperforms the strongest baseline across all three metrics. On Reuters, it achieves the best ACC and NMI while remaining competitive on ARI.

- **Robustness against Missing Views:** In the challenging scenario with 50% missing data, HAMC maintains its superiority on almost all datasets. This validates the robustness of our asymmetric view alignment mechanism, which ensures that reliable views unidirectionally guide unreliable ones to prevent noise propagation. While HAMC ranks second on Caltech-101, the difference is minimal with a gap of only 0.4% in ACC compared to CANDY, suggesting that our consensus-aware learning strategy can construct robust global patterns

even under severe information incompleteness.

Overall, these extensive comparisons confirm that the proposed HAMC achieves state-of-the-art performance in both accuracy and robustness.

### 4.3. Visualization

To verify whether our model can effectively distinguish between reliable and unreliable views, we visualize the entropy distribution of the cluster assignments on the NoisyMNIST dataset in Figure 4. NoisyMNIST is characterized by having one clean view of original digits and one noisy view. As shown in the histogram, the model produces a clear distinction between the two views. View 0 is concentrated in the lower entropy region, indicating that the model assigns high confidence to this view, correctly identifying it as the informative view. In contrast, View 1 exhibits a distribu-

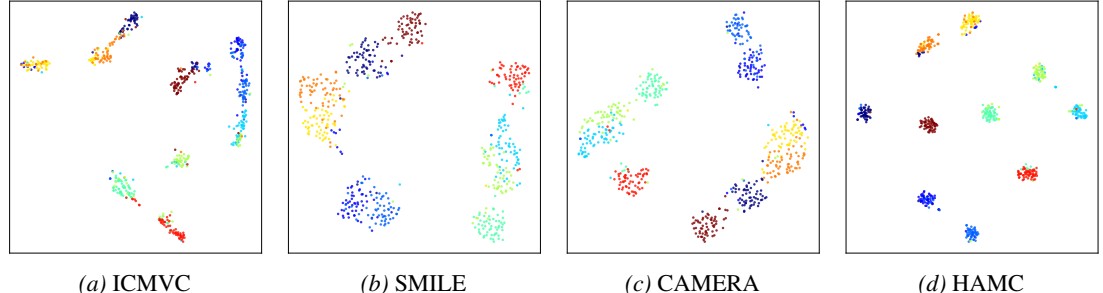

*(a)* ICMVC      *(b)* SMILE      *(c)* CAMERA      *(d)* HAMC

*Figure 3.* $t$-SNE (Maaten & Hinton, 2008) visualization on CUB dataset.

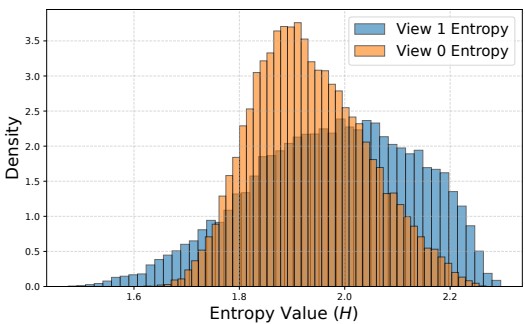

*Figure 4.* Entropy distribution on the NoisyMNIST dataset.

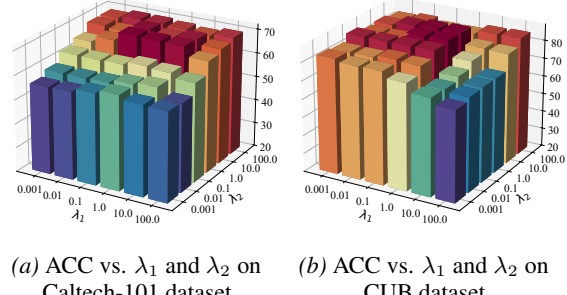

*(a)* ACC vs. $\lambda_1$ and $\lambda_2$ on Caltech-101 dataset      *(b)* ACC vs. $\lambda_1$ and $\lambda_2$ on CUB dataset

*Figure 5.* Sensitivity studies of HAMC on the hyper-parameters $\lambda_1$ and $\lambda_2$.

tion shifted toward higher entropy, reflecting the uncertainty caused by the heavy noise. This evidence strongly supports our asymmetric view alignment strategy: the high-quality view acts as a teacher to guide the noisy view, preventing noise propagation and preserving discriminative features.

To intuitively evaluate the quality of the learned representations, we visualize the feature distributions using t-SNE (Maaten & Hinton, 2008) on the CUB dataset. Figure 3 compares our proposed HAMC against three competitive baselines: ICMVC, SMILE, and CAMERA. Standard t-SNE is commonly applied to Euclidean feature vectors by default. Therefore, we visualize the tangent feature $\mathbf{h}$ instead of hyperbolic embeddings $\mathbf{z}$. While baselines (Figure 3a-3c) exhibit blurred boundaries and overlapping clusters, HAMC (Figure 3d) demonstrates superior intra-class compactness and inter-class separability. This tangent-space separability is consistent with the learned hyperbolic structure, since the exponential map preserves local angular information. The radial relation between the Poincaré ball and the tangent space suggests that samples assigned larger radial distances can also exhibit larger tangent-space norms under the inverse mapping. This provides a possible geometric explanation for the improved visual separation in Figure 3d. Since t-SNE is a nonlinear visualization method, these plots should be interpreted as qualitative evidence rather than a direct proof of hyperbolic separability.

### 4.4. Ablation Studies and Parameter Analysis

To strictly evaluate the contribution of each component in HAMC, we compare the full model with four variants: (1) w/o Hyperbolic: replaces the Poincaré ball model with Euclidean geometry; (2) w/o Asymmetric Align: reverts to rigid symmetric alignment; (3) w/o Uncertainty Weight: employs equal weights for view decision fusion instead of uncertainty-aware weight $\alpha_i^{(v)}$; and (4) w/o Confidence Mask: removes the confidence-based screening mechanism. The experimental results on six benchmark datasets are reported in Table 2. Overall, HAMC achieves the strongest performance among these variants across the reported datasets and metrics, suggesting that the proposed components generally contribute to the final performance. Specifically, the w/o Hyperbolic variant suffers a significant performance drop on most datasets, demonstrating that flat Euclidean space is insufficient to capture the hierarchical uncertainty of multi-view data. Furthermore, removing the asymmetric alignment mechanism leads to noticeable degradation, particularly on the CUB dataset. This confirms that symmetric alignment forces reliable views to align with corrupted ones, propagating noise across modalities. Finally, the w/o Confidence Mask variant exhibits the most severe decline on Caltech-101, suggesting that filtering out ambiguous samples is crucial for preventing error accumulation during self-training.

To better illustrate the stability of HAMC, we investigate the sensitivity of HAMC to the hyper-parameters $\lambda_1$ and $\lambda_2$ on Caltech-101 and CUB datasets. We perform a grid search by varying both parameters within the range of $\{10^{-3}, 10^{-2}, 10^{-1}, 1, 10, 100\}$. The ACC results are visualized in Figure 5. As shown in the chart, HAMC maintains robust performance across a reasonable range of parameter settings. When $\lambda_1$ and $\lambda_2$ are selected from the ranges of $\{0.1, 1\}$, HAMC can achieve desirable performance for both Caltech-101 and CUB datasets.

## 5. Conclusion

In this paper, we presented the HAMC framework to address the mismatch between complex multi-view data and the embedding geometry. Unlike traditional methods that rely on flat Euclidean space, our approach leverages the Poincaré ball model to capture the varying uncertainty of data. HAMC improves cluster separability by using radial confidence modeling in the Poincaré ball, which encourages high-confidence samples to have larger radial distances and reduces the influence of ambiguous or noisy samples. Furthermore, we overcome the limitation of rigid alignment by introducing an asymmetric view alignment mechanism. This mechanism enables reliable views to guide unreliable ones, effectively mitigating the impact of corrupted data. Finally, a consensus-aware learning strategy with a confidence-based screening scheme is designed to refine the cluster structure. Extensive experiments on six benchmark datasets demonstrated that HAMC achieves state-of-the-art performance in both accuracy and robustness. In future work, we intend to investigate adaptive curvature learning, enabling the model to dynamically adjust the manifold geometry to the intrinsic structure of diverse datasets.

## Software and Data

The code repository is available at: [https://github.com/wangemm/HAMC-ICML2026](https://github.com/wangemm/HAMC-ICML2026).

## Impact Statement

This paper presents work whose goal is to advance the field of Machine Learning. There are many potential societal consequences of our work, none of which we feel must be specifically highlighted here.

## Acknowledgements

This work was supported in part by the National Natural Science Foundation of China under Grant 62402235, 62276143 and 62272035, the Beijing Natural Science Foundation (L252202), and the Basic Research Program of Jiangsu (BK20253006).

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

# A. Appendix

## A.1. Notation Summary

Table 3 summarizes the key notations used in this paper.

*Table 3.* Summary of notations used in HAMC.

| Notations | Meaning |
|---|---|
| $\mathcal{X}$ | The multi-view dataset |
| $N$ | The number of samples |
| $M$ | The number of views |
| $K$ | The number of clusters |
| $B$ | The number of samples in a mini-batch |
| $d$ | The dimension of the hyperbolic embedding |
| $\mathbb{D}^d$ | The Poincaré ball model of hyperbolic space |
| $\mathcal{T}_{\mathbf{x}}\mathbb{D}^d$ | The tangent space at point $\mathbf{x}$ |
| $\mathbf{x}_i^{(v)}$ | The input data of the $i$-th sample in view $v$ |
| $\hat{\mathbf{x}}_i^{(v)}$ | The reconstructed input data of the $i$-th sample in view $v$ |
| $\mathbf{h}_i^{(v)}$ | The tangent embedding of the $i$-th sample in view $v$ |
| $\mathbf{z}_i^{(v)}$ | The hyperbolic embedding of the $i$-th sample in view $v$ |
| $\boldsymbol{\mu}_k$ | The learned cluster prototypes |
| $\theta^{(v)}, \phi^{(v)}$ | The parameters of the encoder and decoder for view $v$ |
| $H_i^{(v)}$ | The prediction entropy of the $i$-th sample in view $v$ |
| $w_i^{u \to v}$ | The asymmetric alignment weight from view $u$ to view $v$ |
| $\alpha_i^{(v)}$ | The uncertainty-aware weight of view $v$ for sample $i$ |
| $\mathbf{Q}^*$ | The global consensus distribution |
| $B_i$ | The binary reliability mask for sample $i$ |
| $\lambda_1, \lambda_2$ | The trade-off hyper-parameters for joint training |
| $\lambda_{wc}$ | The weighting coefficient for the warm-up stage |
| $\tau, \tau_c$ | The temperature parameters |
| $m$ | The momentum coefficient for prototype update |
| $p$ | The confidence-based screening ratio |
| $\mathcal{L}_{rec}$ | The reconstruction loss |
| $\mathcal{L}_{align}$ | The asymmetric alignment loss |
| $\mathcal{L}_{cluster}$ | The clustering loss |
| $\mathcal{L}_{warmup}$ | The warm-up objective function |
| $\mathcal{L}_{total}$ | The total objective function |

## A.2. Algorithm

Algorithm 1 summarizes the training process of the proposed HAMC.

---

**Algorithm 1** Optimization of the proposed HAMC

---

**Input:** Multi-view dataset $\mathcal{X}$, Number of clusters $K$, Hyper-parameters $\lambda_1, \lambda_2, \tau, \tau_c, m, p$, Max epochs $T_{max}$, Warm-up epochs $T_{warm}$.
**Output:** Clustering partition result $Y$.
**Initialization:** Randomly initialize the learnable parameters $\theta, \phi$ of autoencoders.
*// Stage 1: Warm-up*
**for** $t = 1$ to $T_{warm}$ **do**
    Update $\theta, \phi$ by minimizing $\mathcal{L}_{warmup}$.
**end for**
Initialize cluster prototypes via K-Means on learned features.
*// Stage 2: Joint Training*
**for** $t = 1$ to $T_{max}$ **do**
    Obtain hyperbolic embeddings $\mathbf{z}_i^{(v)}$ for all views and compute reconstruction loss $\mathcal{L}_{rec}$ via Eq. 5.
    Calculate prediction entropy $H_i^{(v)}$ and asymmetric weights $w_i^{u \to v}$.
    Compute alignment loss $\mathcal{L}_{align}$ via Eq. 8.
    Calculate view weights $\alpha_i^{(v)}$ and fuse global consensus $\mathbf{Q}^*$.
    Generate pseudo-labels $y_i^*$ and reliability mask $B_i$.
    Compute clustering loss $\mathcal{L}_{cluster}$ via Eq. 11.
    Calculate total loss $\mathcal{L}_{total}$ and update network parameters.
    Update cluster prototypes via momentum strategy.
**end for**
**Return:** Final cluster assignments $Y$ based on $\mathbf{Q}^*$.

---

## A.3. Experiments

### A.3.1. DETAILS OF DATASETS

The experiments are conducted on six multi-view datasets: Caltech-101, CUB, LandUse-21, NoisyMNIST, Reuters, and Scene-15. Detailed descriptions are provided below.

- **Caltech-101**: This dataset contains 8677 instances from 101 categories. Following (Zeng et al., 2024), deep features extracted by DECAF (Krizhevsky et al., 2017) and VGG19 (Simonyan & Zisserman, 2015) are used as two views.

- **CUB**: This dataset includes several categories of birds. Following (Li et al., 2025), visual features extracted by GoogLeNet (Szegedy et al., 2015) and text features extracted by doc2vec (Le & Mikolov, 2014) are used as two views.

- **LandUse-21**: This dataset is a remote sensing dataset with 2,100 images belonging to 21 classes. We employ PHOG and LBP features as two views following (Guo et al., 2024).

- **NoisyMNIST**: This dataset is a variant of the classic MNIST dataset. It includes a clean view and a noisy view corrupted by white Gaussian noise. Following (Zeng et al., 2024), we randomly select 30,000 instances for evaluation since some baselines cannot deal with such a large-scale dataset.

- **Reuters**: This dataset is a multilingual news dataset containing 18,758 samples from 6 categories. Following (Guo et al., 2024), English and French are used as two different views.

- **Scene-15**: This dataset comprises 4,485 images classified into 15 scene categories. Following (Guo et al., 2024), PHOG and GIST are employed as two distinct views.

### A.3.2. ADDITIONAL EXPERIMENTAL RESULTS

**Convergence Analysis.** To verify the training stability of HAMC, we visualize the changes in objective loss and clustering performance during the training process. Figure 6 displays the convergence curves on Caltech-101 and CUB datasets. As shown in the figure, the training process exhibits a stable pattern. In the initial epochs, the total loss decreases rapidly. Simultaneously, the clustering metrics show a sharp upward trend. This indicates that the model effectively learns useful features and cluster structures in the early epochs. As the training continues, both the loss and the evaluation metrics gradually become stable with little fluctuation. The curves clearly show that the decrease in loss corresponds to the

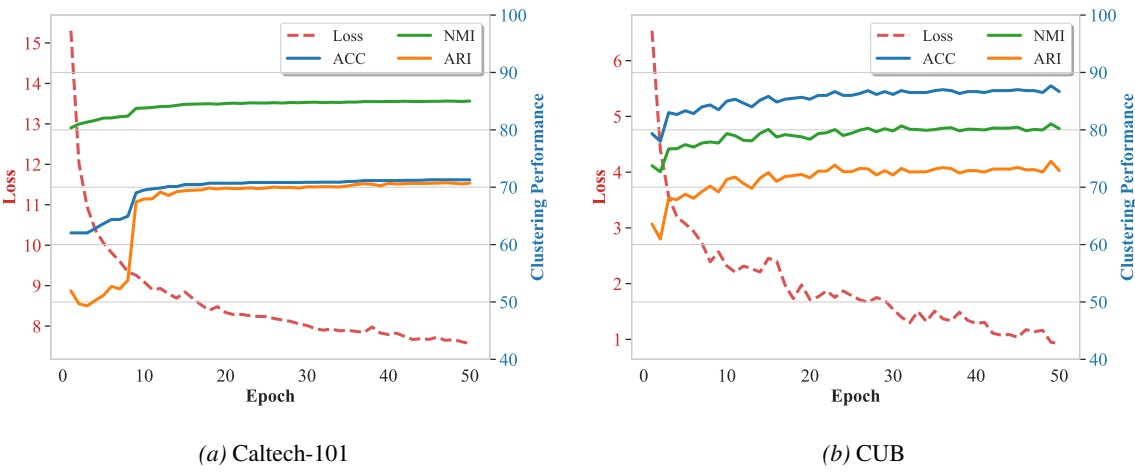

*(a)* Caltech-101            *(b)* CUB

*Figure 6.* Convergence analysis of HAMC on Caltech-101 and CUB datasets. The red dashed line represents the total training loss on the left axis, while the solid lines denote the clustering metrics including ACC, NMI, and ARI on the right axis.

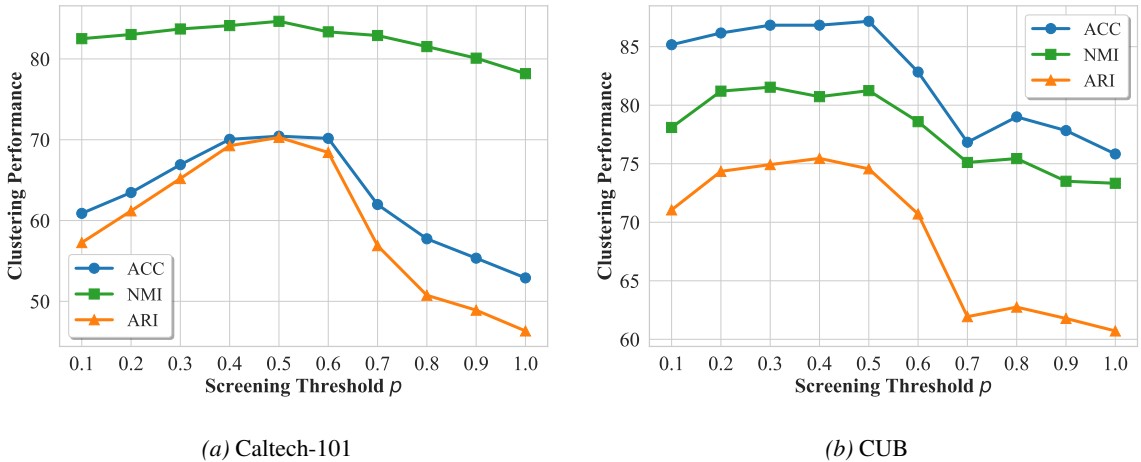

*(a)* Caltech-101            *(b)* CUB

*Figure 7.* Sensitivity analysis of the confidence-based screening ratio $p$ on Caltech-101 and CUB datasets.

improvement in clustering performance. These results confirm that our method has good convergence properties and can be optimized efficiently.

**Sensitivity analysis of ratio $p$.** We further investigate the impact of the confidence-based screening ratio $p$ on the clustering performance. Figure 7 reports the results on Caltech-101 and CUB datasets. As observed, the performance improves as $p$ increases from a small value. This suggests that including more high-confidence samples helps the model learn better cluster structures. However, the performance drops noticeably when $p$ becomes too large. This is because an overly high ratio allows noisy samples with incorrect pseudo-labels to mislead the model optimization. The results demonstrate that setting $p$ to a moderate value (e.g., $p = 0.5$) achieves the best balance between sample quantity and reliability, validating the effectiveness of our screening mechanism.

