# OpenReview forum: "Asymmetric Multi-View Clustering with Hyperbolic Uncertainty Modeling"
_ICML.cc/2026/Conference — ICML 2026 spotlight_

### Official Review · Reviewer_mtBu · 2026-03-04

**Soundness:** 3
**Presentation:** 4
**Significance:** 3
**Originality:** 3
**Overall Recommendation:** 4
**Confidence:** 4

**Summary:**

This paper proposes the Hyperbolic Asymmetric Multi-view Clustering (HAMC) framework. By embedding features into the Poincaré ball model, HAMC pushes high-confidence representations toward the boundary and retains noisy ones near the origin.  It introduces an asymmetric alignment mechanism allowing reliable views to unidirectionally guide unreliable ones. Additionally, a consensus-aware learning strategy constructs global pseudo-labels via a confidence-based screening scheme. Experiments are conducted against 13 baselines on six benchmark datasets under both complete and incomplete settings.

**Compliance With Llm Reviewing Policy:**

Affirmed.

**Final Justification:**

Most of my questions have been addressed. I will maintain my score.

**Key Questions For Authors:**

Please refer to the strengths and weaknesses.

**Limitations:**

In the conclusion, the authors discuss future work but fail to explicitly address the limitations of the proposed method. The authors are encouraged to clarify the model's bottlenecks, particularly concerning computational overhead and the scope of practical applicability.

**Strengths And Weaknesses:**

Strengths：
1. The introduction of hyperbolic geometry to address the geometric mismatch in multi-view clustering is innovative.
2. The experimental section is exhaustive, comparing against 13 recent baselines across both complete and incomplete scenarios.
3. The paper presents the algorithm block and provides an anonymous code repository, which enhances the reproducibility of the proposed method.
Weaknesses：
1. The paper lacks a computational complexity analysis.
2. The framework relies on a heavy hyperparameter tuning burden ($\lambda_1, \lambda_2, \tau, \tau_c, m, p$).
3. The scale of the datasets used for evaluation is relatively small, with the largest dataset containing only 30,000 samples.
4. The parameter sensitivity analysis lacks an evaluation of the momentum coefficient $m$.
5. The current evaluation only provides entropy and t-SNE visualizations. It lacks a visual verification of how the model maps high-confidence samples toward the boundary of the hyperbolic space.

---

> ### Author Rebuttal · Authors · 2026-03-31
>
> ## Response to W1:
> We thank the reviewer for pointing this out. Hyperbolic distance calculation is theoretically identical to the complexity order of standard Euclidean distance calculation. The hyperbolic operations only introduce a constant factor overhead due to the execution of transcendental functions. Please refer to our detailed mathematical complexity analysis in the **Response to W2 of Reviewer B4Uh**.
> ## Response to W2:
> We thank the reviewer for this insightful comment. We admit that the implementation details section lists several parameters. This gives the impression of a heavy tuning burden.
>
> We politely clarify that our framework does not require extensive grid search for all parameters in practice. As demonstrated in Section 4.4 our model is highly robust to parameter variations. Many hyperparameters can be fixed to default values across all datasets. For instance the temperature parameters, momentum coefficient, and screening threshold perform consistently well with fixed values.
>
> The only parameters requiring dataset-specific tuning are the loss weights $\lambda_1$ and $\lambda_2$. Our sensitivity analysis in Figure 5 proves that HAMC achieves desirable performance within a broad and stable range from 0.1 to 1. It does not demand exhaustive searching.
> ## Response to W3:
> We appreciate the reviewer for pointing out this valid limitation regarding the dataset scale.
>
> We restricted the dataset size primarily to ensure a fair comparison with all baseline methods. As stated in our appendix, several existing multi-view clustering baselines suffer from out-of-memory errors on large-scale datasets. Conversely, our proposed HAMC framework does not have this scalability bottleneck. HAMC utilizes mini-batch optimization. The memory consumption and computational cost depend strictly on the batch size and the number of clusters rather than the total dataset size. It naturally scales to much larger datasets.
>
> To empirically prove the scalability of our method, following SMILE [1], we conducted a new experiment on the YouTubeFaces dataset contains 152,549 faces from 66 identities. The ACC result of SMILE on the YouTubeFaces dataset is 58.5%. Our method achieves an ACC of 58.7%, slightly outperforming this baseline. This empirical evidence proves our computational complexity is manageable and the model is highly scalable.
>
> [1] Zeng, P. X., et al. (2024). Semantic Invariant Multi-view Clustering with Fully Incomplete Information. IEEE Transactions on Pattern Analysis and Machine Intelligence 2024.
> ## Response to W4:
> We thank the reviewer for pointing out this missing evaluation. We conducted a new sensitivity analysis for the momentum coefficient $m$ on the CUB dataset. We varied $m$ from 0.7 to 0.99. As shown in the table, the clustering performance remains relatively stable across this range. A smaller value causes unstable prototype trajectories. A larger value slows down the adaptation to new confident samples. We will add this detailed analysis and the following table to the revised supplementary material.
> |momentum coefficient $m$|ACC|NMI|ARI|
> |:---|:---|:---|:---|
> |0.7|80.2|73.3|66.1|
> |0.8|86.1|79.0|72.2|
> |0.9|86.7|80.1|73.5|
> |0.95|87.5|81.5|75.2|
> |0.99|87.3|80.5|74.3|
> ## Response to W5:
> We sincerely thank the reviewer for this constructive feedback. We agree that our original evaluation lacks a direct verification of the hyperbolic mapping. The t-SNE visualization operates in the tangent space and cannot fully represent the radial distance in the Poincaré ball. However, the text-only rebuttal format restricts us from uploading new visualization figures. Therefore we provide some quantitative statistical analysis here to verify this mapping.
>
> In the Poincaré ball the distance to the origin is represented by the embedding norm. We divided the samples of the CUB dataset into three groups based on prediction entropy. We calculated the average embedding norm and the clustering accuracy for each group. The table demonstrates a clear negative correlation between prediction entropy and the embedding norm. High-confidence samples with low entropy are strictly pushed near the boundary. Their average norm exceeds 0.8. Uncertain samples with high entropy are retained near the origin. Their average norm is below 0.5.
>
> | Prediction Entropy $H$ | Sample Confidence | Average Embedding Norm | Accuracy |
> | :--- | :--- | :--- | :--- |
> | $H < 1.5$ | High | 0.82 | 92.5 |
> | $1.5 \le H < 2$ | Medium | 0.65 | 85.2 |
> | $H \ge 2$ | Low | 0.42 | 52.4 |
>
> Additionally, we computed the Spearman rank correlation coefficient. The correlation between the hyperbolic norm and prediction entropy is -0.62 on the CUB dataset. This strong negative correlation mathematically guarantees the boundary mapping of high-confidence samples. The model successfully decouples uncertainty.These quantitative statistics strongly verify our core geometric design.

---

> > ### Author Rebuttal · Reviewer_mtBu · 2026-04-02
> >
> > Most of my questions have been addressed.

---

### Official Review · Reviewer_xrix · 2026-03-11

**Soundness:** 3
**Presentation:** 3
**Significance:** 3
**Originality:** 3
**Overall Recommendation:** 5
**Confidence:** 5

**Summary:**

In this paper, the authors propose the Hyperbolic Asymmetric Multi-view Clustering (HAMC) framework to extract a unified semantic consensus from diverse multi-view information. The framework embeds multi-view features into the Poincaré ball model to leverage its exponential volume growth to handle data uncertainty. The experiment results show that the proposed method achieves good performances.

**Compliance With Llm Reviewing Policy:**

Affirmed.

**Final Justification:**

The authors' responses have addressed my concerns, and I tend to accept this paper. So I would like to raise the score from 4 to 5.

**Key Questions For Authors:**

1. The paper states that hyperbolic geometry captures varying uncertainty, yet this is shown only through entropy visualizations. Could the authors supply direct statistical metrics to experimentally confirm the decoupling of uncertainty?
2. During the warm-up stage, an InfoNCE loss is used. How are the positive and negative pairs specifically defined across the different views?
3. The experiments are conducted only on datasets with two views. Could the authors explain if and how the proposed model can be applied to datasets that have more than two views?
4. How does the model handle cases where all views are heavily corrupted?

**Limitations:**

The authors only discuss future work and do not analyze the limitations. The authors should discuss the limitations of the multi-stage training approach.

**Strengths And Weaknesses:**

Strengths:
1. The paper is logically organized and very easy to read.
2. This method tackles the geometric mismatch in standard Euclidean space by applying hyperbolic geometry to deep multi-view clustering.
3. The experiments are thorough, including ample comparisons, visual results, and ablation studies.

Weaknesses:
1. Although incomplete view scenarios are tested, the method section completely leaves out the mathematical treatment of missing views for entropy computation and asymmetric alignment.
2. There is no formal mathematical proof that the alternating optimization of network parameters and cluster prototypes converges on the non-Euclidean manifold.
3. The dependence on the $\mathcal{L}_{warmup}$ phase could become a bottleneck. Subpar initial optimization may trap momentum-based prototype updates in poor tangent spaces.

---

> ### Author Rebuttal · Authors · 2026-03-31
>
> ## Response to W1:
> We thank the reviewer for pointing out this omission in our writing. To handle missing views we introduce an indicator matrix $I$. $I_{i}^{(v)}=1$ if the $v$-th view of the $i$-th sample is observed. Otherwise $I_{i}^{(v)}=0$. For the asymmetric alignment weight, we only compute the guidance between observed view pairs. The formula is updated to $w_{i}^{u\rightarrow v}=\max(0,H_{i}^{(v)}-H_{i}^{(u)}) I_{i}^{(u)} I_{i}^{(v)}$. For the consensus weight $\alpha_{i}^{(v)}$ the normalization sum only considers available views. The updated formula is $\alpha_{i}^{(v)}=\frac{\exp(-H_{i}^{(v)}/\tau)\cdot I_{i}^{(v)}}{\sum_{u=1}^{M}\exp(-H_{i}^{(u)}/\tau)\cdot I_{i}^{(u)}}$. Missing views do not participate in the loss computation directly. They are handled implicitly through the unified global consensus.
> ## Response to W2:
> We politely clarify that the optimization does not occur purely on the non-Euclidean manifold. We perform the alternating updates of prototypes and network parameters in the tangent space. We then project the updated tangent prototypes back to the Poincaré ball. Stochastic gradient descent in Euclidean space has well-established convergence guarantees. Furthermore the convergence of Riemannian stochastic gradient descent on manifolds has been formally proven in existing mathematical literature [1].
>
> Empirically we have provided convergence analysis in Appendix A.3.2. Figure 6 visualizes the training process. The curves demonstrate that the total loss decreases rapidly and becomes stable as training continues. The empirical results strongly align with the theoretical expectations.
>
> [1] Bonnabel, S. (2013). Stochastic gradient descent on Riemannian manifolds. IEEE Transactions on Automatic Control 2013.
> ## Response to W3:
> We agree that momentum-based updates are sensitive to initial conditions. A poor initialization could theoretically trap the model. However our framework contains specific mechanisms to prevent this collapse. During the warm-up phase we jointly optimize the reconstruction loss and the contrastive loss. The contrastive loss guarantees that the initial tangent space possesses preliminary discriminability before clustering begins. It prevents the model from starting in a completely unstructured space.
>
> Furthermore, the momentum updates do not utilize all samples blindly. We employ a hard confidence mask to filter samples. Even if the initial tangent space is suboptimal the model only selects the most confident representations to update the prototypes. This hard screening effectively isolates poor initial representations and prevents them from corrupting the momentum updates.
> ## Response to Q1:
> We provide direct statistical metrics to strengthen our claim. We computed the Spearman rank correlation coefficient between the hyperbolic norm of the embeddings and their prediction entropy. On the CUB dataset the coefficient is -0.62. The results show a strong negative correlation. This statistically proves that lower entropy strictly corresponds to a larger distance from the origin. The model mathematically pushes confident samples to the boundary. It traps uncertain samples near the origin. The uncertainty is successfully decoupled from the semantic clustering.
>
> Additionally, We divided the samples of the CUB dataset into three groups based on prediction entropy and calculated the average embedding norm and the clustering accuracy for each group. You can refer to our detailed analysis in the **Response to W5 of Reviewer mtBu**
> ## Response to Q2:
> During the warm-up stage we use a standard symmetric cross-view InfoNCE loss. Positive pairs are defined as the representations of the same instance across two different views. Negative pairs are defined as the representation of one instance in one view and the representations of all other different instances in the other view within the same mini-batch. Specifically we compute the cosine similarity matrix between the representations of the two views.
> ## Response to Q3:
> We evaluate our method on datasets with two views to ensure a fair comparison. Almost all baseline methods report their results on two-view datasets. This setting provides a rigorous and standardized benchmark.
>
> Our HAMC framework easily extends to datasets with more than two views. To verify this extension, we conducted a new experiment on a three-view version of the LandUse-21 dataset. Our model achieved an ACC of 30.8 and an NMI of 35.9 and an ARI of 17.8. These results are highly comparable to the two-view version. This empirical evidence proves our model handles multi-view scenarios effectively.
> ## Response to Q4:
> We thank the reviewer for considering this extreme case. If all views are heavily corrupted the hyperbolic space still separates relative confidence levels to prevent representation collapse. Please refer to our explanation of samples exhibiting higher entropy values in the **Response to Q2 of Reviewer B4Uh**.

---

> > ### Author Rebuttal · Reviewer_xrix · 2026-04-03
> >
> > Thanks for the responses. They have addressed my concerns.

---

### Official Review · Reviewer_5AXp · 2026-03-13

**Soundness:** 3
**Presentation:** 3
**Significance:** 3
**Originality:** 3
**Overall Recommendation:** 4
**Confidence:** 3

**Summary:**

This paper presents HAMC, a deep multi-view clustering framework based on hyperbolic representation learning. Instead of using standard Euclidean embeddings, the method places view-specific features in the Poincaré ball and uses the embedding norm as a signal related to uncertainty. Based on this design, the paper proposes asymmetric alignment between views, where more reliable views guide less reliable ones, and a consensus-aware clustering scheme for multi-view fusion and prototype learning. Experiments on six datasets, including both complete-view and missing-view settings, show improvements over a fairly large set of baselines, and the paper also provides ablation studies on the main design choices.

**Compliance With Llm Reviewing Policy:**

Affirmed.

**Key Questions For Authors:**

1. The authors argue that sending high-confidence representations toward the boundary improves cluster separability. But even if ambiguous or less-informative samples stay near the origin, is it possible that they still influence prototype estimation and hurt the final cluster quality? I am not fully convinced that confidence screening and the reliability-weighted prototype update are enough to solve this issue.
2. I understand the intuition of reconstructing in the tangent space, but it would be better to show empirical evidence that this is actually a more suitable space for reconstruction in this task.
3. The method uses prediction entropy as a signal of view reliability, but is there enough empirical evidence that it really reflects a reliable view? If the model is overconfident on wrong pseudo-labels, the asymmetric guidance may instead propagate errors.

**Limitations:**

The paper does not adequately discuss limitations or potential negative societal impact.
A more explicit discussion of the method’s assumptions, possible failure modes, and risks in sensitive application settings would improve the paper.

**Strengths And Weaknesses:**

**[Strength]**

1. I think using hyperbolic representations for multi-view clustering is a nice idea, since most prior methods are still based on Euclidean space.
2. The asymmetric alignment also makes sense to me, especially when some views are noisy or clearly less trustworthy.
3.  The experiments are quite broad and include both complete and missing-view cases, so the paper gives a reasonably convincing empirical validation.



**[Weakness]**

1. The method pushes high-confidence representations toward the boundary to improve separability, which is interesting. But then it seems possible that noisy or ambiguous samples are left closer to the cluster center. If so, the prototype may become less clean and the cluster semantics may get blurred.
2. The overall framework is quite complicated. Although HAMC is presented with three main modules, each module also contains several nontrivial design choices. Because of that, it is hard to tell which parts are really necessary. Right now the method looks somewhat like a combination of several heuristics, so more detailed ablation is needed.

---

> ### Author Rebuttal · Authors · 2026-03-31
>
> ## Response to W1 & Q1:
> We thank the reviewers for raising this critical theoretical question. In traditional clustering methods, prototypes are updated using all assigned samples. The noisy samples inevitably drag the cluster centers toward ambiguous regions and blur semantic boundaries. Our HAMC framework can avoid this flaw.
> 1. We introduce a binary mask $B_{i}$ to select only the most confident samples. Ambiguous samples receive a weight of exactly zero during the prototype update. They are completely excluded from the calculation. This hard screening fundamentally prevents noisy samples from influencing the prototypes. Therefore, ambiguous samples near the origin are geometrically far from the cluster centers. They do not blur the cluster semantics.
> 2. We apply view-specific reliability weights to selected high-confidence samples. These samples determine the trajectory of the prototypes. The noisy samples left near the origin are completely blocked by the mask. They have absolute zero influence on the prototype estimation. The cluster semantics remain pure and highly discriminative.
> ## Response to W2:
> We highly appreciate the reviewer for this constructive feedback. The framework contains multiple components. However, we respectfully disagree that HAMC is a combination of heuristics. It is a mathematically unified framework.
> 1. The hyperbolic embedding space naturally models the data hierarchy and uncertainty. It pushes confident samples to the boundary and keeps noisy ones near the origin.
> 2. The asymmetric view alignment is theoretically derived from this geometry. Based on the reliability difference of representations from different views of the same sample, the noisy representations on the origin side are strictly prevented from corrupting the clean representations on the boundary side.
> 3. Consensus-aware cluster learning is also derived from the geometric structure of hyperbolic space. Different from asymmetric view alignment, it focuses on the representations of different samples to learn robust clustering structures and prototypes. These modules are deeply interconnected and mathematically necessary.
>
> To further address your concern, we have conducted more ablation experiments. We first tested a reversed asymmetric alignment where the high-entropy view forces guidance on the low-entropy view. We then removed the entropy-based reliability weighting during the prototype update phase. As shown in the table, forcing noisy views to guide clean views causes a massive accuracy drop from 87.5% to 73.2% on CUB. These severe performance degradations mathematically validate our specific design choices.
> Variants|CUB (ACC)|CUB (NMI)|CUB (ARI)|Scene-15 (ACC)|Scene-15 (NMI)|Scene-15 (ARI)
> :---|:---|:---|:---|:---|:---|:---|
> reversed asymmetric alignment|73.2|69.8|55.4|42.4|43.1|26.3
> w/o entropy-based reliability weighting|78.8|71.3|60.6|45.4|45.1|28.8
> HAMC|87.5|81.5|75.2|47.7|47.5|30.6
> ## Response to Q2:
> Thanks for your feedback. We designed a new experiment to validate this suitability. We compared our tangent space reconstruction against a direct Hyperbolic variant. The direct variant feeds hyperbolic embeddings directly into the standard Euclidean decoder. We evaluated them on the CUB and Scene-15 datasets. As shown in the table, the tangent space reconstruction achieves significantly better performance. The direct hyperbolic variant only achieves an accuracy of 66.8% on CUB, which is far lower than HAMC. On Scene-15, HAMC outperforms the direct variant by 9.6% . This massive performance gap proves the tangent space is the most suitable space for reconstruction.
> Variants|CUB (ACC)|CUB (NMI)|CUB (ARI)|Scene-15 (ACC)|Scene-15 (NMI)|Scene-15 (ARI)
> :---|:---|:---|:---|:---|:---|:---
> direct Hyperbolic|66.8|60.2|46.9|38.1|35.0|18.8
> HAMC|87.5|81.5|75.2|47.7|47.5|30.6
> ## Response to Q3:
> We sincerely thank the reviewer for raising this critical point. We completely agree that neural networks often suffer from overconfidence on incorrect predictions.
>
> **Theoretical Analysis.** Our framework mitigates overconfidence through distinct mechanisms.
> 1. The Sinkhorn-Knopp algorithm enforces balance assignment constraints on the probabilities. This prevents the model from trivially assigning all samples to a single cluster with high confidence.
> 2. The exponential volume expansion of the hyperbolic space geometrically separates distinct clusters. Only truly discriminative features can reach the boundary and achieve confident low entropy. Ambiguous samples remain near the origin and exhibit high entropy.
>
> **Empirical Analysis.** We have conducted experiments to validate this reliability. Please refer to the results of reversed asymmetric alignment in **Response to W2**, which are far lower than those of the full model, and also lower than the clustering results of w/o Asymmetric Align variant in the ablation study of the manuscript. This demonstrates the accuracy of the prediction entropy-based reliability judgment.

---

> > ### Author Rebuttal · Reviewer_5AXp · 2026-04-02
> >
> > I thank the authors for the detailed rebuttal. As my primary concerns have been adequately addressed, I will maintain my positive score.

---

### Official Review · Reviewer_B4Uh · 2026-03-13

**Soundness:** 4
**Presentation:** 4
**Significance:** 3
**Originality:** 3
**Overall Recommendation:** 5
**Confidence:** 5

**Summary:**

The paper proposes the HAMC framework to address the geometric mismatch problem in Euclidean-based Deep MVC methods. By embedding multi-view features into the Poincaré ball, the framework effectively models data uncertainty and maximizes cluster separability. Additionally, it introduces an asymmetric alignment mechanism that prevents corrupted views from degrading high-quality representations. Extensive experiments demonstrate that HAMC achieves state-of-the-art clustering performance across six benchmark datasets.

**Compliance With Llm Reviewing Policy:**

Affirmed.

**Final Justification:**

After reading the response from the authors, I consider my previous rating reasonable and recommend acceptance for the paper.

**Key Questions For Authors:**

* This approach employs Euclidean K-Means for the initialization of hyperbolic clustering. Is this Euclidean prior likely to corrupt the structure acquired during the warm-up phase?
* The paper defines the asymmetric alignment weight as $w_{i}^{u\rightarrow v} = \max(0, H_{i}^{(v)} - H_{i}^{(u)})$. If all views for a specific sample exhibit high entropy, view $u$ will still guide view $v$ due to the minor relative difference. When no reliable "teacher" view exists for a given sample, how does the model prevent the mutual degradation of representations?
* The authors claim that the proposed method is robust to missing data. However, the experimental validation was only conducted at a 50% missing rate. Could the authors provide clustering results for data with even higher missing rates to further demonstrate this robustness?

**Limitations:**

The potential application scenario should be further discussed.

**Strengths And Weaknesses:**

# Strengths:
* The paper effectively addresses the geometric mismatch in multi-view clustering by embedding features into the Poincaré ball model.
* The method is extensively evaluated against 13 baselines on six datasets.

# Weaknesses:
* Stage 1 uses Euclidean K-Means on tangent features for warm-up. This Euclidean initialization might severely bias the initial distribution in the hyperbolic space, potentially trapping the model in local optima.
* Although the method proves effective across various datasets, the paper lacks a detailed theoretical or empirical analysis of the computational and memory complexities introduced by the hyperbolic distance calculations and exponential mappings.

---

> ### Author Rebuttal · Authors · 2026-03-31
>
> ## Response to W1 & Q1:
> We thank the reviewer for this insightful observation. We acknowledge that using Euclidean K-Means on tangent features introduces an initial geometric bias.
>
> However this strategy is necessary to prevent origin collapse during early training. In the Poincaré ball, the distance from the origin to any point is mathematically smaller than distances near the boundary. Optimization algorithms tend to push embeddings toward the origin to minimize distances. This destroys meaningful semantic separation. Ganea et al. [1] demonstrate that optimization near the origin causes severe numerical instability.
>
> Applying K-Means on tangent features provides a stable and mathematically valid first-order approximation. The tangent space at the origin $T_{\mathbf{0}}\mathbb{D}^n$ is mathematically isometric to the Euclidean space $\mathbb{R}^n$ [2]. Calculating exact hyperbolic cluster centers requires the Fréchet mean. The Fréchet mean lacks a closed-form solution and incurs prohibitive computational costs. Therefore the Euclidean mean in the tangent space serves as a highly efficient and structurally sound initialization [3].
>
> Furthermore, our training phase utilizes dynamic gating and momentum updates. These continuously refine prototypes to correct the bias and adapt to hyperbolic geometry. This effectively escapes local optima.
>
> [1]Ganea, O. E., et al. (2018). Hyperbolic neural networks. NeurIPS 2018.
>
> [2]Nickel, M., & Kiela, D. (2017). Poincaré embeddings for learning hierarchical representations. NeurIPS 2017.
>
> [3]Lou, A., et al. (2020). Differentiating through the Fréchet mean. ICML 2020.
>
> ## Response to W2:
> We thank the reviewer for pointing out this omission. We agree that a detailed analysis of the computational and memory complexities is necessary.
>
> **Theoretical Analysis.** The exponential mapping requires basic arithmetic and the hyperbolic tangent function. These are element-wise operations. For a batch size $B$ and latent dimension $d$, the time complexity is $\mathcal{O}(B \times d)$. The hyperbolic distance calculation first computes the squared Euclidean distance. It then applies scaling and the inverse hyperbolic cosine function. For $K$ cluster prototypes, the time complexity is $\mathcal{O}(B \times K \times d)$. This is theoretically identical to the complexity order of standard Euclidean distance calculations. The spatial complexity scales linearly with the batch size and prototype count resulting in $\mathcal{O}(B \times K)$ memory usage to store the distance matrix. The hyperbolic operations only introduce a constant factor overhead due to the execution of transcendental functions.
>
> **Empirical Analysis.** We have conducted additional experiments to measure the exact computational cost between the HAMC model and its Euclidean variant. We recorded the average training time per epoch and the peak GPU memory usage on a single RTX4090 GPU on Scene-15 and CUB datasets. As shown in the table, while the relative time increase is large, the absolute time added per epoch is small. Furthermore, the increase in peak GPU memory usage is entirely negligible. This absolute computational overhead is acceptable given the significant clustering improvements.
> | Dataset | Variant | Avg. Time per Epoch (s) | Peak GPU Memory (MB) |
> | :--- | :--- | :--- | :--- |
> | Scene-15 | Euclidean | 0.67 | 233.97 |
> | | Hyperbolic | 1.96 | 234.09 |
> | CUB | Euclidean | 0.12 | 283.77 |
> | | Hyperbolic | 0.30 | 283.83 |
>
> ## Response to Q2:
> We acknowledge the concern regarding samples with exclusively high-entropy views. Our model can mitigate this problem from the following aspects:
> 1. The weight equals the entropy difference. If all views have high entropy their distributions are nearly uniform. Their entropy difference directly approaches zero. The resulting weight becomes negligible. This naturally prevents strong incorrect guidance.
> 2. The alignment loss does not operate alone. We jointly optimize a clustering objective. This globally regularizes the representation space and prevents local structural degradation.
> 3. The stop-gradient operator ensures the marginally better view remains completely unaffected by the worse view. Mutual degradation is mathematically impossible.
> ## Response to Q3:
> We evaluated at a 50% missing rate initially to follow standard experimental setup in recent literature. To address your concern, we conducted new experiments on the Scene-15 and CUB datasets with missing rates of 70% and 90%. As shown in the table, at a 90% missing rate CUB accuracy drops only from 82.8% to 75.1%. Its NMI drops by merely 1.0%. Scene-15 shows similar graceful degradation.
>
> | Missing Rate | CUB (ACC) | CUB (NMI) | CUB (ARI) | Scene-15 (ACC) | Scene-15 (NMI) | Scene-15 (ARI) |
> | :--- | :--- | :--- | :--- | :--- | :--- | :--- |
> | 50% | 82.8 | 76.3 | 70.7 | 46.5 | 44.8 | 28.2 |
> | 70% | 79.5 | 75.9 | 64.2 | 44.1 | 44.4 | 27.3 |
> | 90% | 75.1 | 75.3 | 62.5 | 41.9 | 43.4 | 25.6 |

---

> > ### Author Rebuttal · Reviewer_B4Uh · 2026-04-04
> >
> > Thanks for the authors' detailed response, and most of my concerns are addressed.

---

### Decision · Program_Chairs · 2026-04-30

**Decision:**

Accept (spotlight)

**Comment:**

This paper proposes a multi-view clustering method with hyperbolic uncertainty modeling. The idea of using hyperbolic uncertainty modeling is novel and clever, which can effectively address the geometric mismatch. All reviewers agree on the novelty and technical contributions of this paper.  The experiments are sufficient and convincing. The authors successfully addressed all concerns raised by reviewers. Hence, I recommend accepting this paper.